The co-evolution of multiply-informed dispersal: information transfer across landscapes from neighbors and immigrants

Chaine Alexis S. alexis.chaine@ecoex-moulis.cnrs.fr 1
Legendre Stéphane 2
Clobert Jean 1
1 Station d’Ecologie Expérimentale du CNRS à Moulis , Moulis , France
2 Laboratoire Ecologie & Evolution , Paris , France
Gelfand Mikhail
Electronic publication date: 2013 Feb 26
Publication date: 2013
Volume: 1
Electronic Location ID: e44
Received 2012 Dec 19; Accepted 2013 Feb 8
Copyright: © 2013 Chaine et al.
Copyright year: 2013
Copyright holder: Chaine et al.
License: This is an open access article distributed under the terms of the Creative Commons Attribution License, which permits unrestricted use, distribution, and reproduction in any medium, provided the original author and source are credited.
License URL: https://creativecommons.org/licenses/by/3.0/

Keywords: Social information, Dispersal, Immigrant-dependent, Meta-population, Density-dependent, Evolution, Adaptive dynamics

Funding: DIAME and INDHET program Biodiversa TenLamas CNRS and ANR-INFO-EVO-ECO ANR-JCJC NetSelect JC was supported by French ANR BLANC grants DIAME and INDHET, and program Biodiversa TenLamas. JC and ASC received support from the CNRS and ANR-INFO-EVO-ECO, and ASC received support from ANR-JCJC NetSelect. This work is part of the Laboratoire d’Excellence (LABEX) entitled TULIP (ANR-10-LABX-41). The funders had no role in study design, data collection and analysis, decision to publish, or preparation of the manuscript.

==============================
Dispersal plays a key role in natural systems by shaping spatial population and evolutionary dynamics. Dispersal has been largely treated as a population process with little attention to individual decisions and the influence of information use on the fitness benefits of dispersal despite clear empirical evidence that dispersal behavior varies among individuals. While information on local density is common, more controversial is the notion that indirect information use can easily evolve. We used an individual-based model to ask under what conditions indirect information use in dispersal will evolve. We modeled indirect information provided by immigrant arrival into a population which should be linked to overall metapopulation density. We also modeled direct information use of density which directly impacts fitness. We show that immigrant-dependent dispersal evolves and does so even when density dependent information is available. Use of two sources of information also provides benefits at the metapopulation level by reducing extinction risk and prolonging the persistence of populations. Our results suggest that use of indirect information in dispersal can evolve under conservative conditions and thus could be widespread.

Introduction

Dispersal is a key component of many ecological and evolutionary processes ranging from population dynamics to local adaptation and has been the focus of extensive empirical and theoretical investigation (Clobert et al., 2001; Ronce, 2007; Nathan et al., 2008; Clobert et al., 2012). The impact of dispersal on population dynamics, movement across the landscape, and local adaptation makes it a critical element of understanding how populations are affected by landscape fragmentation and global warming (Chaine & Clobert, 2012). Dispersal has largely been treated as a population level character even though dispersal decisions are fundamentally an individual behavior that should benefit from knowledge of the landscape. Recent empirical evidence suggests that information use in making dispersal decisions and navigating the landscape plays an important role in patterns of dispersal (Bowler & Benton, 2005; Clobert et al., 2009; Schmidt, Dall & Van Gils, 2010). Information use would cause a shift in how we view dispersal. Exchanges among populations would no longer represent a random subset of genotypes and might affect local adaptation patterns (Blanchet, Clobert & Danchin, 2010). Dispersers might not spread randomly across the landscape and some populations might receive more or fewer immigrants due to dispersal costs (Bonte et al., 2012). In applied work, if we want to encourage dispersal, we would need to make sure that the key information sources are available or even manipulate information to get the desired level of dispersal (Blanchet, Clobert & Danchin, 2010; Chaine & Clobert, 2012). Yet our fundamental understanding of informed dispersal remains limited (Clobert et al., 2009).

The use of information in dispersal decisions has received attention through a limited range of possibilities despite potentially important effects on fitness (Ims & Hjermann, 2001; Ronce et al., 2001; Bowler & Benton, 2005; Clobert et al., 2009; Schmidt, Dall & Van Gils, 2010). Classical ecological (metapopulation) and evolutionary (gene-flow) theory assumes constant dispersal rates with random movement and no information use (Hanski & Gaggiotti, 2004). At the other extreme, ideal free settlement models assume perfect knowledge of the entire landscape which influences dispersal (Holt & Barfield, 2001). Both approaches are analytically tractable, but biologically unrealistic since organisms often use some information (Greene, 1987; Danchin et al., 2004; Dall et al., 2005; Avarguès-Weber, Dawson & Chittka, 2013) but rarely have perfect information. Significant progress in understanding dispersal itself will require specific attention to biologically plausible mechanisms for gathering information (Schmidt, Dall & Van Gils, 2010).

Recent models have investigated how information on local population density affects dispersal (Travis, Murrell & Dytham, 1999; Cadet et al., 2003; Ronce, 2007; Enfjäll & Leimar, 2009; Hovestadt, Kubisch & Poethke, 2010; Bocedi, Heinonen & Travis, 2012), but it is becoming increasingly clear that organisms use a variety of information sources (Ronce et al., 2001; Danchin et al., 2004; Bonnie & Earley, 2007; Clobert et al., 2009; Schmidt, Dall & Van Gils, 2010) that might inform them about the presence or content of other populations in the landscape without direct measurement. We call these forms of information that do not result from direct sampling of the environment “indirect information” (Doligez, Danchin & Clobert, 2002; Danchin et al., 2004; Blanchet, Clobert & Danchin, 2010). For example, tourists in Paris are easily identified by the fact that they are using maps (unlike Parisians) and this might suggest to Parisians that there is indeed a habitable world outside of Paris. These more “indirect” sources of information derived from the observation of conspecifics are more controversial because they less accurately predict fitness in any given patch (Schmidt, Dall & Van Gils, 2010). However, indirect information carries a distinct advantage of providing some information about other patches without requiring costly exploration of other sites.

A few recent empirical examples in birds, lizards, and other organisms now suggest that indirect social information is accessible and used by individuals in making dispersal decisions (Doligez, Danchin & Clobert, 2002; Cote & Clobert, 2007a; Chaine et al., 2010; De Meester & Bonte, 2010). For example, in the common lizard (Zootoca vivipara), juveniles use a number of direct and indirect sources of information to make dispersal decisions (Clobert, Massot & Le Galliard, 2012). Juveniles gain direct information by sampling the density of their patch (Le Galliard, Ferriere & Clobert, 2003) and regarding kin competition (especially mother-offspring competition; Léna et al., 1998; De Fraipont et al., 2000). However, juveniles also gain indirect information based on the arrival of new immigrants (Cote & Clobert, 2007b; Cote, Boudsocq & Clobert, 2008; Cote & Clobert, 2012) and the failure of emigrants to find new populations (Cote & Clobert, 2007a). Likewise, some spider species, use both direct sampling of information on density, habitat quality, and wind direction (Bonte et al., 2003a; Bonte et al., 2003b; Bonte, Bossuyt & Lens, 2007; Bonte, Van Belle & Maelfait, 2007; De Meester & Bonte, 2010) as well as indirect information such as the number of other individuals dispersing (De Meester & Bonte, 2010). Yet it remains unclear how prevalent use of indirect information in dispersal might be across species. Widespread use of indirect information would dramatically alter our understanding of dispersal and would have consequences for both fundamental work in ecology and evolution as well as applied conservation.

Using a theoretical model, we show that simple rules for the use of indirect social information in dispersal decisions can evolve under a broad range of conditions and therefore might be quite common in nature. We investigated the evolution of information use prior to dispersal using a simple metapopulation model in which we allowed information use in dispersal to evolve. We were primarily interested in whether the use of indirect information provided by immigrants could evolve, and if so, could it evolve in competition with direct information about local density.

The model

We constructed an individual-based model of informed dispersal behavior, based on information about the local density and/or the number of immigrants, while simplifying the landscape and genetic features of the system. This individual-based model follows a female-based life cycle with two age-classes (individuals in the population are juveniles from birth until age 1, subadults from age 1 to 2, and adults after age 2 and age-specific survival and fecundity (Caswell, 2001, and see Fig. 1). Only juveniles dispersed and this dispersal depended on baseline uninformed dispersal that alleviates kin competition (U) and informed dispersal (D and I) as described below. Our basic model used a “fast” life history roughly equivalent to a small lizard or passerine life cycle (survival: s0 = 0.2, s1 = 0.35, s2 = 0.5; fecundity: f1 = 7, f2 = 7, see Schoener et al., 2003; Legendre et al., 2008). In each patch, discrete time structured population dynamics were modeled. Juveniles were given the opportunity to disperse to other patches prior to the subsequent reproductive episode if they survived their first year. All patches were equally connected (leading to lower kin competition) and population size was limited at reproduction by the maximum patch carrying capacity which was the same for all patches (K = 100). This configuration leads to very stable populations with low levels of demographic stochasticity, lower kin competition, and very small benefits of dispersal (populations are all similarly near K) essentially creating a conservative scenario for the evolution of informed dispersal. Subsequent simulations introduced increased stochasticity to explore the benefits of information use under other scenarios (see SOM).

Figure 1 Life cycle of organisms in the model.

Diagram of the basic life cycle of individuals in the model. The two age classes of reproductive individuals (subadults aged 1 year, and adults aged 2 years and more) are described by their age-specific survival (s) and fecundity (f). Individuals disperse during the juvenile stage from age 0 to 1, indicated by *.

Basic loop

The simulation is in discrete time. Individuals are described by their age, the values of their adaptive traits, their patch of residence, their dispersal status, the strategy they played if they dispersed, the probability of dispersal, and the cost of dispersal.

At each time step, the following operations are performed:

Survival

Reproduction and mutation

Dispersal

Increase time step

(1) Survival: Surviving juveniles become subadults, surviving subadults become adults, and adults have a constant survival rate. Survival was determined by a Bernoulli draw according to the age-specific survival.

(2) Reproduction: Subadults and adults reproduced according to their age-specific reproductive rate. Fecundity was drawn using a Poisson distribution, but limited by the patch carrying capacity.

(3) Mutation: Offspring inherited their parental dispersal genotype (coefficients of the dispersal functions, D and I, described below) with a 0.02 probability of mutation. The degree of mutation on D, I, and U in later models (see additional results in SOM) was set by a random draw from a Gaussian distribution with a standard deviation of 0.02. These mutations have the effect of causing a slight alteration in how intensely the dispersal decision will respond to a given set of local cues (local density and number of immigrants).

(4) Dispersal: Offspring were given the opportunity to disperse according to their dispersal strategy (i.e. genotype), and current conditions that informed their dispersal strategy. Specific dispersal strategy functions are described below. Since the first individuals to disperse at a given time step would only have access to local density information (no immigrants possible since nobody has yet dispersed), we randomized at each time step the order in which individuals were selected for dispersal across the whole metapopulation. We chose to model dispersal behavior on current density and immigrant number rather that everyone using the same values from the previous time step because it reflects a much more biologically realistic mechanism for information use in dispersal as newborns gather information about their surroundings (Matthysen, 2012). If an individual juvenile dispersed, it could die during dispersal according to the costs of dispersal (varied in simulations from 0–0.1 in additional results; see SOM) or arrive at a new destination patch. This cost of dispersal modified the juvenile survival rate (s0 × (1-cost)). Juveniles who survived dispersal, were randomly assigned a new patch, excluding their natal patch, and were then counted as an immigrant for that new patch.

Initiation of the simulation began with the creation of 100 subadult individuals with identical genotype in a single patch. Individuals then reproduced and their offspring who dispersed began to colonize the patches.

Dispersal functions

We modeled two forms of information use that could influence dispersal: (1) a measure of the local density which are known to provide a benefit to dispersal behavior (Cadet et al., 2003) and (2) a measure of the number of immigrants entering a patch (Cote & Clobert, 2007a). Local density directly influences reproductive success whereas the number of arriving immigrants indicates that other populations are attainable and may provide some information about overall metapopulation density. The influence of local density and immigrant-borne information on dispersal behavior (Bx) were modeled as: (1) Density-dependent:BD=DniKi−2

(2) Immigrant-dependent:BI=IMi−2

where ni is the number of individuals in patch i, Ki is the patch carrying capacity K, and Mi is the number of immigrants entering the patch. The coefficients (D and I) influenced the intensity of these behaviors and each was free to evolve independently of the other. Immigrant-dependent dispersal only occurred if immigrants were present (i.e. if Mi > 0). Fixed intercepts (−2) were included to set a lower limit to dispersal via each form of information use at 12%. This intercept allowed dispersal to evolve more rapidly without having an impact on the evolved dispersal rate which was always significantly higher (see SOM, Fig. S8). We assumed haploid genetics and clonal reproduction with mutation in “genes” for the coefficients (D and I) that affect each informed-dispersal strategy as described above. These behaviors were then used to determine the probability of dispersal, d(x), associated with density (dD) and immigrant (dI) information sources using the following function: (3) d(x)=11+exp(−x)

where x is the influence of each form of information described by Eqs. (1) and (2) (x = BD or BI). This function allowed us to convert the biologically meaningful relationships described in Eqs. (1) and (2) to probabilities of dispersal dD and dI respectively.

Because immigrant-dependent dispersal can only occur if immigrants exist (i.e. some dispersal already occurs), we also included a fixed parameter for baseline uninformed dispersal (dU = 0.1). Uninformed dispersal should alleviate kin competition and increased values when allowed to evolve would be favored when kin competition is higher. Dispersal was always drawn for uninformed dispersal first (dU) and then for informed dispersal (dD or dI). Removal of this baseline dispersal prevents the evolution of immigrant-dependent dispersal when alone (I-only models) since there were no immigrants, but it had little influence on the evolution of density-dependent dispersal (D-only models) or both density and immigrant-dependent dispersal when both were present (D&I models; Fig. S9). Allowing this baseline uninformed dispersal (U) to evolve had little effect on the evolution of informed dispersal (D-only, I-only, or D&I; Fig. S10).

We constructed alternative models of information use to examine the independent effects of density (D-only) or immigrants (I-only) on dispersal as well as their joint co-evolutionary dynamics when individuals could use both forms of information simultaneously (D&I). In models including both density- and immigrant-dependent information (D&I), all individuals were capable of using both sources of information and the sum of the two sources of information determined the dispersal probability. This assumption matches empirical findings that individuals use multiple sources of information in decision making (Le Galliard, Ferriere & Clobert, 2003; Cote & Clobert, 2007b; Cote, Boudsocq & Clobert, 2008; Clobert et al., 2009; Clobert, Massot & Le Galliard, 2012; Cote & Clobert, 2012; Matthysen, 2012). In the case of simultaneous models the individual dispersed with probability dD + dI if dD + dI < 1, and always dispersed if dD + dI > 1. We calculated the “realized” informed dispersal rates attributed to each information source by a random draw using the relative dispersal probability of each information source (dD or dI). The probabilities dD and dI represent the incentive of an individual to disperse according to density or immigrant information, and are not the realized dispersal rates associated with each strategy. These realized dispersal rates were computed as the total number of individuals dispersing according to each strategy divided by the total number of individuals in the metapopulation.

We determined the probability that informed dispersal evolved and the dispersal rate associated with information use using Monte Carlo simulations of 100 trajectories over 1.5 × 106 time steps for each set of parameters and each model case. Because all individuals were capable of information use from one or two sources, then all values of the evolved coefficient potentially existed in the population unless the entire metapopulation went extinct. Therefore, we determined that “evolution” of an informed dispersal strategy had occurred if the evolved coefficient was greater than 0 more often than by chance across simulations since drift should lead to negative coefficients as often as positive ones. This approach gives similar results to quantifying evolution if it increases above an estimate of random drift as presented in the supplemental materials (see SOM).

Our initial model exploration focused on the use of density and immigrant sources of information and the coevolution of both forms when together. Subsequent models (see SOM) explored the effects of variation in life history, carrying capacity, patch number, environmental stochasticity, the costs of dispersal (Bonte et al., 2012), variation in baseline dispersal (dU), the order in which different sources of information are used, and the immigrant information use strategy function.

Results

Evolution of information use: single source of information

We found that informed dispersal could evolve and drive dispersal behavior and metapopulation dynamics under a broad range of contexts. Consistent with other models (Travis, Murrell & Dytham, 1999; Ronce, 2007), we found that density dependent dispersal evolves when it is the only source of information (Figs. 2A, 3). Here we show that the arrival of immigrants also provides useful information that can drive dispersal behavior (Figs. 2B, 3). Indeed, information-dependent dispersal coefficients (D and I) were significantly biased towards positive values in contrast to expectations from drift which should lead to an equal probability of positive and negative values (Sign test: D-only: 97/100 positive trials, P < 0.0001; I-only: 99/100 positive trials, P < 0.0001). Both density and immigrant dependent dispersal evolved even when each was in competition with uninformed dispersal (fixed dU = 10% and when U was allowed to evolve; see SOM and Figs. S9, S10a) and lead to increased dispersal from that source of information (Figs. 2A, B) despite a highly stable and homogenous landscape. Dispersal reaction norms due to information use illustrate this nicely: local density and immigrant number influence dispersal (Fig. 4A, B respectively) at equilibrium compared to a flat, fixed dispersal rate of uninformed dispersal. Density-dependent dispersal shows a steady increase in dispersal as local density rises (Fig. 4A). In contrast, immigrant-dependent dispersal shows a rapid increase in dispersal with the first few immigrants and then quickly asymptotes at high levels of dispersal (Fig. 4B).

Figure 2 Temporal dynamics of the evolution of informed dispersal.

Temporal dynamics of the evolution of information based dispersal due to local density (dD in red) and the number of arriving immigrants (dI in blue). Trajectories reflect average dispersal rates for 100 Monte Carlo simulations. (A) Dynamics of immigrant number information use alone (dI). (B) Dynamics of density dependent information use alone (dD). (C) Dynamics of both density dependent and immigrant dependent information when used simultaneously (D&I) with no cost of dispersal. Uninformed dispersal is fixed at 10% and does not evolve. The 95% confidence interval is shown for the last time step on each trajectory.

For informed dispersal to evolve there must be some benefit to these strategies. Individuals benefit from dispersal when they find a new population with a lower density given that fitness is density-dependent. We compared the density of the new destination patch and an individual’s original patch right before reproduction to estimate the benefit of dispersal to that individual. Informed dispersal led to discovery of a less dense patch than the population of origin on average. Both density and immigrant information seemed to present very similar advantages early in the evolutionary process (Fig. S5a, b and Fig. 5A). However, the benefit of informed dispersal was extremely slight (0.5%–0.02%) since the landscape was largely homogenous and most populations were very close to their carrying capacity at all times. Environmental stochasticity augmented spatial heterogeneity in patch density and led to a larger benefit during the evolution of informed dispersal (Fig. 5A; Fig. S5; see also McPeek & Holt, 1992; Travis & Dytham, 1999).

Evolution of information use: multiple sources of information

Coexistence of density and immigrant dependent dispersal occurred often in our model when both forms of information use were possible (48% of simulations for model D&I; Figs. 2C and 3). Information-dependent dispersal coefficients for both behaviors (D and I) were again significantly biased towards positive values overall in contrast to expectations from drift (Sign test for D&I model: D: 65/100 positive trials, P = 0.035; I: 82/100 positive trials, P < 0.0001). Reaction norms of density- and immigrant-dependent dispersal both show increases with density or immigrant number respectively and rise well above background levels of uninformed dispersal (Fig. 4C, D). If we contrast these reaction norms to the reaction norms that evolve when only one form of information use is possible, we see that the slope of density dependent dispersal decreases considerably (Fig. 4A vs. C) whereas the shape of the immigrant-dependent dispersal curve changes only slightly (Fig. 4B vs. D; dispersal above 98% at 3 vs. 5 immigrants respectively). Optimal levels of density-dependent dispersal therefore shift considerably when another source of information affects dispersal. In contrast, immigrant-dependent information has large effects on dispersal with the arrival of the first few immigrants and this trigger does not change much when other sources of information are available.

Figure 3 Evolution of information use.

Probability that each form of information use evolves. Plotted are the proportion of simulations where dispersal evolved based on density dependent information (D, red), immigrant information (I, blue), both density and immigrant information (D + I, red and blue hatch), or where dispersal did not evolve (None, white) across 100 Monte Carlo simulations. D-alone and I-alone are for models with just one source of information available (plus U fixed at 10%). D&I is a model with both density and immigrant dependent dispersal present.

Figure 4 Behavioral reaction norms of informed dispersal.

Reaction norms for informed dispersal behavior. Solid lines show the reaction norms (black) and 95% CL (grey) for each form of dispersal. Dashed lines reflect uninformed baseline dispersal. Reaction norms were created using the Informed Dispersal equations with the mean evolved coefficient after 100000 generations. Lines for the 95% CL were constructed using the variance in evolved coefficients among 100 Monte Carlo runs. Top panels are for models where only one source of information is possible and show dispersal due to (A) density dependent dispersal (D-only) and (B) immigrant dependent dispersal (I-only). Bottom panels are for models where only both sources of information are possible (D&I) and show dispersal due to (C) density dependent dispersal and (D) immigrant dependent dispersal.

Figure 5 Benefits of informed dispersal.

The relative benefit of dispersal behavior to an individual is estimated by how much better a disperser did by moving (i.e. old pop density/new pop density, both at reproduction). Shown is the dispersal benefit over the first 100000 time steps for models with low environmental stochasticity (5% of populations hit) in models (A) I-only (D-only is similar) or (B) D&I. Benefits of multiply-informed dispersal (D&I) relative to using no information or a single source of information (D or I-only) is also observed at the meta-population level by reducing global extinction risk (proportion of 100 Monte Carlo simulations where the metapopulation goes extinct) as stochasticity increases due to (C) random environmental stochasticity or (D) small population size.

While both forms of dispersal evolved less often when both were present (a decrease of 32% and 17% for density and immigrant dependent dispersal respectively), coexistence remained high when in competition with a second source of information (D&I) relative to models where just one strategy was possible (D-only or I-only; Fig. 3). Joint evolution of both information use behaviors occurred even in competition with uninformed baseline dispersal (see SOM, Figs. S9, S10, S11).

Informed dispersal showed benefits at the metapopulation level when both forms of information were used together relative to using just one source of information. This benefit was most apparent when demographic stochasticity increased. Lower population carrying capacities raised the risk of extinction due to increased demographic stochasticity, and for a narrow window of carrying capacities the use of two sources of information helped reduce the risk of extinction for the metapopulation as a whole by 20%–40% relative to use of just one source of information (Fig. 5D). At slightly lower carrying capacities, when metapopulation extinction always occurred, the use of two different sources of information lead to longer persistence (200–10000 time steps or roughly 100–5000 generations; Fig. S3) of the metapopulation than if just one source of information was used. An increase in the frequency of environmental stochasticity lead to higher metapopulation extinction, and the risk of extinction was lower when one or more sources of information was available (D-only or I-only or D&I) compared to uninformed dispersal only (U-only) (Fig. 5C).

Discussion and Synthesis

Our results show that informed dispersal evolves under a broad array of contexts and that both density and indirect immigrant-dependent information sources evolve and can coexist. The frequent evolution of informed dispersal in the very conservative setup examined here (e.g. stable metapopulation) suggests that use of a variety of information sources, including indirect measures of the metapopulation landscape, could be common in nature. Indeed, direct information use in dispersal decisions is widespread (Ims & Hjermann, 2001; Matthysen, 2005; Ronce, 2007; Clobert et al., 2009; Schmidt, Dall & Van Gils, 2010) and the few empirical investigations of indirect information use that we are aware of have found evidence for it despite a broad taxonomic range. For example, common lizards modify their dispersal behavior in response to immigrants who appear to provide information about the density of their natal population (Cote & Clobert, 2007a). Likewise, our recent work in Tetrahymena ciliates shows that residents alter their dispersal rate when arriving immigrants come from populations that differ in density or social structure. In both of these empirical examples, immigrants carry more information (e.g. population density) than we included in our model. This additional information should serve to increase the fitness benefits of immigrant-dependent dispersal suggesting that we have probably underestimated the likelihood that it evolves.

For use of both information sources to evolve, there must be benefits to adjust behavior using two sources of information rather than a single source. Benefits of density-dependent dispersal are well known since movement out of high density patches should have direct fitness benefits when reproduction is density dependent (Travis, Murrell & Dytham, 1999; Matthysen, 2005; Cote, Boudsocq & Clobert, 2008). Our results demonstrate that even under very conservative conditions, immigrant dependent dispersal also presents a benefit and evolves. Likewise, coexistence of density- and immigrant-dependent dispersal even under the stable meta-population structure that we modeled suggests that these behaviors can evolve and coexist frequently even when the benefits of each behavior are low. Coexistence also implies that neither source of information carries benefits that would cause competitive exclusion of the other information source. Using two sources of information also provided additional benefits and could play an important role in metapopulation stability, especially as increased stochasticity creates larger inequalities in population densities. The benefits of using multiple information sources (decreased extinction risk) that we measured occurred for a small range of meta-population conditions (medium levels of stochasticity), but they suggest an important advantage to informed dispersal under less stable conditions of most real meta-populations. While the benefits we measured in our model were small in the relatively homogenous landscape we constructed, conditions that more realistically imitate empirical landscapes should confer much larger benefits to this behavior.

Joint evolution of density and immigrant dependent dispersal would be prevented if information content of density and immigrant number were not sufficiently different or if one information source was superior to the other (Enfjäll & Leimar, 2009; Hovestadt, Kubisch & Poethke, 2010; Schmidt, Dall & Van Gils, 2010; Bocedi, Heinonen & Travis, 2012). Immigrant arrival might be related to the overall density of the metapopulation since populations that have more individuals will generate more dispersers, and therefore immigrants, even through a fixed baseline dispersal rate. This estimate of the overall metapopulation density contrasts to density-dependent measures of the local population alone. Competitive exclusion might be expected since immigrant number should be more decoupled with local fitness in any single patch and thus dispersal should carry a higher variance in benefits relative to direct information on local density. In simulations where we introduced a difference in the cost of using each form of information (Figs. S6, S7), the most costly form of information did not evolve – although this cost is not directly linked to the quality of information. More frequently, we found coexistence of information use through both density and immigrant information. This suggests that each source of information is not fully redundant and that one source of information is not necessarily superior to the other. This equivalency of information can serve as an advantage under some contexts (e.g. when stochasticity is high; Figs. S5 and S3) and would be especially useful where the costs of information use from one source might constrain dispersal below an optimal level (Bocedi, Heinonen & Travis, 2012). Likewise, if immigrants also carry additional information about their populations (Cote & Clobert, 2007a) or help orient dispersers towards certain populations, then we could expect the benefits of indirect information use to be even more advantageous.

The potential prevalence of informed dispersal has a number of important implications for both fundamental and applied ecology. In basic ecological research, the use of information has recently been explored in terms of density dependent dispersal, and this simple behavior greatly effects how movement influences population persistence (Ims & Hjermann, 2001; Cadet et al., 2003; Matthysen, 2005). Earlier models of “informed” dispersal – such as “ideal free distribution” models – generally assumed perfect knowledge of the landscape (Abrahams, 1986; Gray & Kennedy, 1994; Holt & Barfield, 2001) which presumably was acquired through prospecting that carried low costs. Low cost prospecting might work when patches are close (e.g. foraging patches), but is less realistic when habitat patches are more distant. The use of indirect information, such as the arrival of immigrants, could provide another mechanism by which the ideal free distribution is achieved (Baguette, Clobert & Schtickzelle, 2010). If immigrant arrival is linked to overall metapopulation density and if immigrants carry additional information about the quality of those habitats as suggested in empirical examples (Cote & Clobert, 2007a), then we might approach an ideal free distribution through use of indirect information transfer across the landscape. Deviation from ideal free models might then in part reflect the quality or reliability of that indirect information transfer (see also Abrahams, 1986; Gray & Kennedy, 1994; Chaine & Clobert, 2012). Most likely, individuals use a number of sources of information on local conditions, direct prospecting of nearby patches, and indirect measures of the landscape such as immigrant-borne information (Clobert et al., 2009). If this form of information use is prevalent, then we must shift our view of dispersal from largely random movement among populations to much more targeted and informed movement patterns that approach ideal-free expectations.

Connectivity and dispersal are crucial aspects of population persistence, yet studies of dispersal and metapopulation dynamics usually ignore the important role that information transfer across the landscape might play in guiding subsequent dispersal decisions. Applied management or conservation efforts to increase connectivity or gene flow might be greatly hampered if we do not also introduce the indirect cues that influence dispersal. Indeed, the highly variable success of artificial corridors (Gilbert-Norton et al., 2010) could in part be caused by the lack of indirect information since immigrants will be rare when a new corridor is first constructed (see also Le Galliard, Ferriere & Clobert, 2003). More generally, conservation efforts could be greatly aided by modifying natural dispersal through the manipulation of information that is accessible to residents rather than by costly alterations of the landscape between habitat patches (Chaine & Clobert, 2012). As we show here, access to multiple sources of information may better mitigate extinction risk in highly stochastic environments compared to situations where little information exists. Broader inclusion of how information is used in dispersal should provide us with new tools for conservation and fundamentally modify our approach to conservation ecology and the management of populations in peril.

Our findings also have important implications for dispersal theory and the incorporation of information use into this field. We found the evolution of both forms of informed dispersal despite potentially large differences in the quality of information gleaned from each source. Whereas local density directly affects fitness, immigrant arrival at best gives some indication of surrounding population sizes when density dependent dispersal exists and at worst simply provides evidence that other populations exist. Coexistence of the two sources of information suggests that the quality of information may be somewhat less important than the presence of that information. In support of this notion, models of indirect information use based on immigrant presence rather than immigrant number show very similar results (Fig. S12). This result is empirically supported by the fact that dispersal in the common lizard was found to be sensitive to the presence and not to the quantity of immigrants (Cote & Clobert, 2007a). Similarly, recent models of density dependent information use suggest that the precision of information provides diminishing returns and high quality information is not optimal when it also incurs elevated costs associated with gathering additional precision (Bocedi, Heinonen & Travis, 2012). Both of these investigations adopt very simple dispersal contexts and yet both show that information use in dispersal evolves quite readily and should be common in nature. More generally, the passive information transfer across the landscape that evolves in our models could be an important first evolutionary step that allows more active information transfer and communication to evolve both within populations and across landscapes.

Supplemental Information

Supplemental Materials Supplemental Information

Click here for additional data file.

We thank O Ronce and B Lyon for comments on an early draft and discussions on this topic and anonymous reviewers for additional comments.

Additional Information and Declarations

Competing Interests

Author Contributions

The authors declare that they have no competing interests. J Clobert is an Academic Editor for PeerJ.

Alexis S. Chaine conceived and designed the experiments, performed the experiments, analyzed the data, wrote the paper.

Stéphane Legendre conceived and designed the experiments, performed the experiments, contributed reagents/materials/analysis tools, wrote the paper.

Jean Clobert conceived and designed the experiments, wrote the paper.

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
