# Peer review of "The co-evolution of multiply-informed dispersal: information transfer across landscapes from neighbors and immigrants"

_PeerJ, doi:10.7717/peerj.44_

## Round 0.1 · original submission · Minor Revisions

In my opinion, the manuscript requires minor revisions before accepting. The reviewer comments are pasted below - please respond to them with your revision.

In addition to those comments, it was also mentioned that the title seems to be cryptic in so far as it does not really reflect the most important message of the paper. I leave it to you whether the title needs changing.

Reviewer 1 ·

Basic reporting

The manuscript is very well written and in general the text is clear and easy to follow. However, there are few instances, where I recommend some modification of the text, some of which appear to me to be a result of some unintended errors in the text.
1. The introduction provides very sound and sufficient background information. However, in my opinion the manuscript would benefit from a more clear definition of “indirect information” (i.e. the number or presence of immigrants) in the introduction. For example, while I found the parallel to tourists in Paris very helpful, it would be nice to make a direct connection to animal populations and describe shortly the main findings of the cited studies (line 55) to clearly transfer to the reader what the authors consider “indirect information” in this study (the same recommendation is valid for the abstract).
2. Line 94 ff: There is some confusion in the description of the formulas used to model the influence of local density and immigrant-borne information on dispersal in the main text. After reading the supplemental online material, I´m convinced that the description in the SOM is correct, while the main text some error occurred. I found the description of the “Basic loop” and the “Dispersal functions” much clearer in the SOM and recommend to simply transfer them into the main text.
3. Line 132-133: “In the case of simultaneous models, if dD + dI > 1, then the individual dispersed, otherwise it dispersed with probability (dD + dI). “
In my opinion the sentence is confusing. I recommend to change the sentence to something like "In case of simultaneous models, individuals dispersed with the probability (dD+dI), and always dispersed if dD+dI>1."
4. Figure 1: I found it difficult understand figure 1 in relation to the text (in the main text as well as in the SOM). Although there are 3 age classes with respective survival rates stated in the text (juvenile, subadult and adult) there are only 2 in the figure. When Yr1 refers to subadult and Yr2 to adult, why the adults survive with the probability s1 (and not s2) in the figure?
I recommend to add some clarification in the text and the use of more intuitive abbreviations in the figure (for example sub=subadult; juv=juvenile etc…)
5. Figure 2: There seems to be a graph missing in figure 2. The text indicates a grap 2D which I could not find in the manuscript.
6. Figure 3: The text according to figure 3 is repeated three times in the manuscript

Experimental design

No comments

Validity of the findings

No comments

Additional comments

The manuscript is very well written and organized, and provides very interesting information on the potential importance of information other than density for decision-making processes and individual dispersal in animals. Like stated by the authors, the results underline the necessity of rethinking the processes involved in dispersal in fragmented landscapes and I´m sure that the manuscript will be an incentive for further investigations of theoretical as well as empirical nature.
I recommend the manuscript for publication with minor revision according to the points mentioned earlier.

Reviewer 2 ·

Basic reporting

No Comments

Experimental design

The Model section needs to be more throughly and clearly written. It would currently be impossible to reimplement the model. See comments in section below.

Validity of the findings

No Comments

Additional comments

Throughout, and especially in the introductory paragraphs, the paper would benefit from drawing from a greater literature for its references. In the first paragraph, for example, a very high proportion of the cited papers are self-cites. This is important not only in terms of having greater representation of the existing, relevant literature but better in terms of the likelihood of this paper being easily found by others. The referencing feels a little lazy at present.

line 36 - what do you mean by 'more or less random movement'? This is too vague!

Line 47-49: to make this analogy entirely clear, I would edit to, ' (unlike Parisians) and this might suggest to Parisians that there is a ...'
Some hyphens would help, especially in the methods. For example, I would sugges ‘discrete-time structured population dynamics were modelled using a two-age class ...’
Lines 80-81. The reason why the scenario you outline here is conservation (in that there is little benefit to dispersal) is because you have global dispersal and thus minimise kin competition as a driving force. This may be obvious to those most familiar with the evolution of dispersal field but it will be less obvious to other likely readers (e.g. from the information ecology field). So I would spell this out clearly hear. I think the same general point is true elsewhere in the paper too. You will hopefully not only have dispersal evolution people reading this, so ensure that you don’t assume too much.

Lines 97 and 99 – there is a mismatch in the symbols used in eqn (2) and as described in the text.
Line 100: I don’t find coefficient I in the equations.

It is impossible to work out how the emigration probability depends on local density, number of emigrants or both together from the methods, as they are currently written. It is not clear how eqns (1) and (2) are integrated with eqn (3). I would suggest that the authors seek a single eqn that expresses the probability that an individual emigrates from its natal patch as a function of both local density and the number of emigrants.

I am concerned with the arbitrary assumption that at least 0.1 individuals disperse (line 118). As I understand it the authors include this such that they initially do have some immigrants into patches in order that selection can operate on the informed dispersal function. But a better way of achieving this would be to simply run the model from different initial conditions including some where there is initially plenty of dispersal. Perhaps there is another justification for this assumption, but the one provided here is not compelling.

This is a discrete time model and the normal method employed is to have all individuals simultaneously assessing local density and then determining if they emigrate. But I think you have serial dispersal events here. However, from the information provided (e.g. lines 119-120: 123-124), I would be unable to recreate what you have implemented. Much more detail is needed.

Lines 212-213 – this is an important result and one that perhaps deserves greater emphasis in the paper., including perhaps in the abstract.

Lines 245-246 (and elsewhere) You need to be a little careful as while most of your simulations were for stable metapopulations, you also present some interesting results where metapopulations are not stable. This currently may lead to some confusion.

Line 254: my guess is that it would only be prevented if there was a cost of acquiring the second source of information (unless the first source was perfect, which most often it won’t be).

---

## Round 0.2 · accepted · Accept

The paper is now accepted, thank you for submitting to PeerJ